# Application of Novel Non-Thermal Physical Technologies to Degrade Mycotoxins

**DOI:** 10.3390/jof7050395

**Published:** 2021-05-19

**Authors:** Mohammad Yousefi, Masoud Aman Mohammadi, Maryam Zabihzadeh Khajavi, Ali Ehsani, Vladimír Scholtz

**Affiliations:** 1Department of Food Science and Technology, Faculty of Nutrition and Food Science, Tabriz University of Medical Sciences, Tabriz 516615731, Iran; yousefi.sbu@gmail.com; 2Student Research Committee, Department of Food Science and Technology, National Nutrition and Food Technology Research Institute, Faculty of Nutrition Sciences, Food Science and Technology, Shahid Beheshti University of Medical Sciences, Tehran 1983963113, Iran; masoud.am70@gmail.com (M.A.M.); mary.khajavi72@gmail.com (M.Z.K.); 3Department of Food Science and Technology, Food and Drug Safety Research Center, Tabriz University of Medical Sciences, Tabriz 516615731, Iran; ehsani@tbzmed.ac.ir; 4Department of Physics and Measurements, University of Chemistry and Technology Prague, Technická 5, 166 28 Prague, Czech Republic

**Keywords:** non-thermal plasma, electron beam, pulsed light, detoxification, filamentous fungi

## Abstract

Mycotoxins cause adverse effects on human health. Therefore, it is of the utmost importance to confront them, particularly in agriculture and food systems. Non-thermal plasma, electron beam radiation, and pulsed light are possible novel non-thermal technologies offering promising results in degrading mycotoxins with potential for practical applications. In this paper, the available publications are reviewed—some of them report efficiency of more than 90%, sometimes almost 100%. The mechanisms of action, advantages, efficacy, limitations, and undesirable effects are reviewed and discussed. The first foretastes of plasma and electron beam application in the industry are in the developing stages, while pulsed light has not been employed in large-scale application yet.

## 1. Introduction

Mycotoxins are a group of low-molecular-weight compounds with a lot of diversity at their structures, which are mainly produced through the secondary metabolism of fungi. They are produced on different types of foods and are considered as hazardous substances for both animal and human health [1]. Their impact on health may be very hard and can be categorized in three forms as mutagenic, carcinogenic, and genotoxic [2,3,4]. On the other hand, the contamination of foodstuffs and plant materials, particularly grains, with mycotoxins goes along with intense financial losses. For example, nearly one third of the total crop value was lost in Hungary in 2014 (around 330 million Euros), partly due to the lowered prices owing to the higher toxin contamination, and partly because of losses in animal husbandry and extra costs of toxin binders, medication, etc. [5]. Therefore, many mycotoxin precautions were established for the pre-harvest stage to the stage after.

Currently, the application of non-thermal techniques has been noticed [6] due to the advantages such as low cost, low time consuming, and low food matrix side effects. Non-thermal plasma (NTP), electron beam (EB) irradiation, and pulsed light (PL) are among the newest and promising technologies used to decrease the concentration of mycotoxins in grains. Moreover, due to the non-thermal character of presented techniques, those impacts on the quality of foods and plants may be, in general, only negligible.

The investigation of parameters affecting the mycotoxins degradation performance of NTP, EB, and PL (Figure 1) is a critical issue discussed among published studies. Therefore, the main purposes of this study involve the introduction of these three techniques and the expression of their applications and their degradation mechanisms against mycotoxins.

## 2. Mycotoxins

Mycotoxins are secondary metabolites produced by filamentous fungi mostly belonging to *Aspergillus*, *Alternaria*, *Penicillium*, and *Fusarium* families and are usually highly toxic [7,8]. Currently, more than 400 mycotoxins produced by about 200 different fungal species have been identified [9]. The main mycotoxins in terms of toxic impact on both animals and humans are aflatoxins, fumonisins, citrinin (CIT), T-2 toxin, cyclopiazonic acid, nivalenol, moniliformin, deoxynivalenol, ochratoxin A, zearalenone, patulin, and ustiloxins [10]. The most frequently encountered mycotoxins in food and agriculture systems are classified into four main groups: (1) aflatoxins, produced by species of *Aspergillus*, and ochratoxins and patulin, produced by *Penicillium* and *Aspergillus* species; (2) fumonisins, zearalenone, and trichothecenes, produced by species of *Fusarium*; (3) ergot alkaloids, produced by species of *Claviceps* [6], and (4) altertoxin, produced by species of *Alternaria* [11].

Mycotoxins can have adverse health effects, since many of them are classified as mutagenic (carcinogenic or genotoxic). Among more than 400 compounds identified as mycotoxins (without masked mycotoxins), 30 have great attention, and they are considered a threat to animal or human health [12]. For example, aflatoxins can harm the immune system through inhibiting the proliferation of cells and protein synthesis [13]. Furthermore, many diseases are accosted with the intake of these toxins. In overdose ingestion, some adverse effects are esophageal cancer, benign endemic nephropathy, liver cancer, equine leuco-encephalo malacia, immunosuppression, hormonal disorders, and even deaths [14]. Mycotoxins have acute or chronic complications. Acute toxicity has a quick onset and a clear toxic response, whereas chronic toxicity is described by low-dose exposure over a long-time stage, leading to cancers and other irreversible impacts. For example, acute aflatoxicosis causes death; while chronic aflatoxicosis leads to immune suppression, cancer, and other slow pathological conditions. The liver is the primary goal organ of aflatoxins. Additionally, fumonisins have been observed as an acute food-borne disease in India, in which the incidence of abdominal pain, borborygmi, and diarrhea was related with the consumption of sorghum and maize contaminated with high concentrations of fumonisins [15].

Contamination of crops and foods by mycotoxins is a serious global problem. Among others, three representative reports follow. BIOMIN [16] company in 2018 reported an average of 40% of agriculture products such as corn, rice, wheat, barley, corn gluten meal, soybean meal, dried distillers grains, and silage had been contaminated by aflatoxins, zearalenone, fumonisins, T-2 toxin, deoxynivalenol, and ochratoxin A [16]. Motloung, De Saeger [17] in 2018 reported that 40% of the 70 food samples of South Africa such as ground chilli, coarse chilli, paprika, ginger, onion spices, chicken spices, beef spices, vegetable spice, Mexican chilli, cheese spices, and fruit chutney spices had been contaminated by aflatoxin B1, aflatoxin G1, sterigmatocystin, ochratoxin A, fumonisin B1, fumonisin B2, acetyldeoxynivalenol, and roquefortine C. Gonçalves, Schatzmayr [18] in 2017 reported that 93 and 72% of 2176 plant and finished aquaculture feed samples in Asia and Europe had been contaminated by aflatoxins (B1, B2, G1, and G2), fumonisins (B1 and B2), ochratoxin A, zearalenone, deoxynivalenol, and T-2 toxin.

The high occurrence of mycotoxins in agriculture and food-based products and their extremely serious health effects on humans and animals have compelled researchers to think of the ways to deal with mycotoxins. Mycotoxins have acute or chronic complications. Acute toxicity has a quick onset and a clear toxic response, whereas chronic toxicity is described by low-dose exposure over a long-time stage, leading to cancers and other irreversible impacts. For example, acute aflatoxicosis causes death; while chronic aflatoxicosis leads to immune suppression, cancer, and other slow pathological conditions. The liver is the primary goal organ of aflatoxins. Additionally, fumonisins have been observed as an acute food-borne disease in India, in which the incidence of abdominal pain, borborygmi, and diarrhea was related with the consumption of sorghum and maize contaminated with high concentrations of fumonisins [15].

Due to the undesirable effect of mycotoxins on health, several approaches have been suggested to avert or degrade the mycotoxin contamination including pre- and post- harvest ones and decontamination. The attention to the operation of pre- and post-harvest agricultural products such as pest control, cultivation techniques, sorting, washing, storing, etc., can meaningfully reduce the number of mycotoxins; however, complete degradation of mycotoxins is impossible by these operations. Thus, other methods are employed to degrade mycotoxins from food, including physical (such as flotation, extraction, thermal treatments, UV, gamma, plasma, and electron beam treatments) [19], chemical (such as alkalization, heating, use of organic acids, and oxidizing compounds) [20], and biological techniques (such as fermentation and microbial enzymatic activities) [21]. Most chemical, microbial, and thermal treatments have several drawbacks such as the removal of some parts of food only, thermal degradation of matrix, consuming a lot of time, or requiring high energy or high cost, using organic solvent, leaving toxic residues, the possibility of adverse reactions of chemical additives with the food matrix, and the possibility of producing unwanted compounds by microorganisms in food [22,23,24]. For these reasons, new approaches such as gamma, pulsed-light, non-thermal plasma, and electron beam treatments have been developed.

## 3. Non-Thermal Plasma NTP

The first presented non-thermal technique is the NTP (denoted also as low-temperature or cold). The term “plasma” denotes a fully or partially ionized gas, primarily consisting of free electrons, ions, and photons as well as atoms in both base and excited states [25,26]. The recent advances of NTP permits the investigations of NTP in several areas such as polymer science [27], microbiology [28], biotechnology [29,30,31,32], or food sciences [33,34]. One of the recently attracted topic is the degradation of mycotoxins, caused by the reactive species of plasma mostly through the oxidation, epoxidation, hydrogenation, cleavage of furan rings, or through the modification of cyclopentanone, lactone ring, or the methoxyl group.

The NTP is typically obtained using an electric discharge or by applying microwaves at atmospheric or reduced pressures. The particular systems designed to produce NTP are the dielectric barrier discharge (DBD) in many modifications, corona discharge, gliding arc discharge, inductively coupled plasma, microwave discharges, or atmospheric pressure plasma jet [35,36,37]. The DBD, as one of the most frequently used systems (Figure 2), consists typically of one or two electrodes placed in a distance of several cm or mm separated by a dielectric plate. A typical high voltage of 10^2^ to 10^4^ V with frequency of 10^0^ to 10^4^ Hz is applied to the electrode between which the plasma is produced by the electric induction. For detail overview see one of several other review papers [31,38,39,40,41,42]. The plasma-chemical processes as particle collisions, ionization, photoemission, photoionization, etc., lead to the production of an array of active species as excited atoms, positive ions, negative electrons, chemical radicals, UV photons, etc. Often, if the air or similar gas is used as the discharge atmosphere, a wide group of reactive oxygen species (ROS) and reactive nitrogen species (RNS) with high microbicidal effects is created. ROS include mainly single delta oxygen, ozone, atomic oxygen, peroxide, and superoxide; RNS consist mainly of nitric oxide (NO), including intermediates, NO radicals, NO^+^, NO^−^, NO_2_, NO_2_^−^, NO_3_^−^, ONOO^−^, N_2_O_3_, and N_2_O_4_ [40,41,42,43]. Compared to other conventional and non-thermal approaches (UV light, pulsed light, and gamma irradiation), NTP acts rapidly against fungi and mycotoxins; whereas, it has a rather milder impact on quality and needs a low energy input.

Plasma technology has gained a special status in many areas of the agriculture system from plant science [44] to the food industry [33]. Concerning plant development, non-thermal plasma can play a substantial role in the seed germination, in increasing the permeability of seeds, in regulation of nitrate metabolism, in activation of photosynthesis, and in a changing the structure of a seed coat [44]. In the food industry, this technology covers a large aspects of foodstuffs, including food microbial decontamination, food allergy mitigation, dissipation of pesticides, modification of food materials, edible oils hydrogenation, packaging substances processing [44], and mycotoxin degradation [41,45,46,47]. As for the last-mentioned item, NTP has big potential to reduce the volume of mycotoxins by both fungi reduction and mycotoxins degradation with high potential for practical applications during the whole food production chain. The most recent studies [48,49,50,51] suggest practical applications of NTP to degrade mycotoxins on roasted coffee, hazelnuts, maize, and rice. The list of these studies can be found in Table 1. Unfortunately, all studies were successful on the laboratory scales only; therefore, more up-scale studies are required.

### 3.1. Mycotoxin-Infected Matrix

A matrix either solid or liquid usually possesses a shielding effect against plasma, thereby reducing its degradation activity. Presumably, matrix components scavenge reactive molecular and atomic species in the plasma, protecting mycotoxins from degradation. As reported by ten Bosch and Pfohl [56], the amount of mycotoxins after NTP treatment remains higher in extracts of rice than that in pure mycotoxins covering the surface of glasses. For example, it took 5 s for fumonisin B1 to be completely degraded from the surface of glass, but this time for inoculated extract of rice with the same amount (100 μg mL^−1^ and estimated thickness of 10 μm) of fumonisin B1 was measured in 60 s. However, in spite of the complex matrix effect, a meaningful decay of mycotoxins also occurred in the matrix.

In a study presented by Ouf and Mohamed [57], they found a reduction of 75–100, 67–92, and 48–78% of mycotoxins in the wash-water of cherries, grapes, and strawberries, respectively, all of them after an exposure to NTP under similar conditions. The variations in the results for different wash-water samples have been attributed to differences in the concentration of antioxidants in the washes leached from fruits through the molding areas or injured skin. The presence of antioxidant substances probably attenuates the produced oxidizing species, mainly oxygen radicals, and consequently decreases the effect of NTP. The wash-water of strawberries seemed to have the highest antioxidant capacity in comparison with grapes and cherries.

### 3.2. Mechanism of Action of NTP against Mycotoxins

Siciliano and Spadaro [58] have found that aflatoxin B1 can be decomposed completely in pure extract, and over more than 70% on dehulled hazelnut by NTP technology. This level of impact depicts the power of non-thermal plasma against mycotoxins, hence indicating the importance of knowing its degradation mechanism. The nature of NTP consists of three agents of heat, UV radiation, and reactive species. The heat (<60 °C) and UV intensity (50 µW/cm^−2^) produced during the generation of non-thermal plasma is far less than needed for mycotoxins degradation. For example, Liu and Chang [59] employed a UV treatment with the intensity of 800 μW cm^−2^ to successfully degrade the AFB_1_ from peanut oil.

Thus, reactive species, including radicals, electrons, ROS, and RNS, are the main factors affecting mycotoxins. In brief, the most important degradation effect of NTP is the quick reaction of the generated reactive spices with functional groups, double and triple bonds, and different active rings related to the structure of mycotoxins. In fact, the breakage of the mycotoxin structure leads to the production of compounds of the lower toxicity compared to the primary mycotoxin. As the concentration of reactive species increases, so does the mycotoxin degradation rate. The increase in the power of the system, in the moisture content of edible materials, or in the time, respectively, could result in the generation of a greater amount of reactive species.

Regarding the action mechanism of these elements on aflatoxin B1, Shi and Cooper [60] have suggested two pathways. These pathways and branches cause the degradation of aflatoxin B1 into six fragmented products. The first degradation pathway (Figure 3a) basically depends on the presence of H_2_O molecules and the neutral, the radical, and the ionized forms of H and CHO, which are produced in the NTP system. The first branch involves hydration, cleavage of methoxy group (OCH_3_), and hydrogenation reactions, where two main products (C_17_H_15_O_7_ and C_16_H_17_O_6_) and one intermediate compound (C_16_H_13_O_6_) are generated. The second branch involves the addition of aldehyde group (OCH) and H_2_, resulting in the generation of C_19_H_15_O_8_ (intermediate product) and C_19_H_19_O_8_ (main fragmented product).

The first pathway reactions occur mainly on the lactone ring and double bond of C8-C9 in the furan ring. This lactone ring, furan ring, and C8-C9 double bond are mainly associated with carcinogenicity, mutagenicity, and teratogenicity of aflatoxins [49]. Therefore, the reactions ensued from NTP generate products of markedly lower toxicity than the primary mycotoxin [50].

The second pathway (Figure 3b) is composed of oxidation and epoxidation reactions. The first branch is related to the formation of C_17_H_13_O_7_ via epoxidation of the aflatoxin B1 terminal double bond by hydroperoxyl radical (HO_2_^•^). The second branch starts with the production of C_14_H_11_O_5_ through the cleavage of aflatoxin B1 furan ring followed by its oxidation by ROS such as OH^•^ and ozone (O_3_) causing the formation of C_14_H_11_O_6_.

Concerning the impact of reactive species produced by NTP on the food quality, there is no sufficient amount of data in order to reach a firm conclusion. However, Feizollahi and Iqdiam [61] showed that NTP is promising technology for barley grains. They reported that NTP treatment for 6 and 10 min decreased deoxynivalenol concentration by 48 and 54%, respectively, while the treatment did not demonstrate significant changes in the quality of grains, including moisture content, protein, and glucan content.

### 3.3. Effective Parameters of NTP on Degradation of Mycotoxins

The mechanism of degradation is not fully understood; the resistance to NTP varies between particular mycotoxins. The recent studies [56,57,58,62,63,64,65,66,67] pertinent to effective parameters of plasma on mycotoxin breakdown have focused on several aspects such as the type of mycotoxin and matrix, the source of plasma, the situations of storage, and the process conditions, including the type of gas, the sample stirring, the time of treatment, the power of system, and the humidity. Some of the most challenging of these parameters are discussed in the following sections.

#### 3.3.1. Structure of Mycotoxin

Based on following publications, it seems that the resistance primary depends on mycotoxin structure and is independent on its molecular mass. The report by ten Bosch and Pfohl [56] implies that the decay rate of enniatin B (681.9 Da) and sterigmatocystin (324.3 Da) in NTP, in spite of differences in the molecular weight, is similar. On the other hand, fumonisin B1 (721.8 Da) and AAL toxin (521.6 Da) had, despite similar masses, the lower half-life in comparison with sterigmatocystin. The same report indicated that toxins containing long aliphatic chains such as fumonisin B1 and AAL have less resistance to plasma compared to toxins with a compact form of aromatic rings such as sterigmatocystin, enniatin B, or zearalenone. Moreover, most other mycotoxins consisting of combined structures of aliphatic chains and condensed rings have an intermediate decay rate.

Other work by Siciliano and Spadaro [58] concerning aflatoxins reported that the NTP sensitivity of aflatoxins B1 and G1 was higher than aflatoxins B2 and G2. The reason for this is attributed to the existence of C8-C9 double bond (olefinic site) within the furan ring of aflatoxins B1 and G1, whilst aflatoxins B2 and G2 lack this double bond. As Jalili reported [68], the oxidative action of NTP constituents, and particularly ozone molecules, causes the opening of the terminal furan ring in this double bond leading to the formation of primary ozonides and their derivatives such as organic acids, aldehydes, and ketones.

#### 3.3.2. Gas, Humidity, Discharge Intensity, and Exposure Time

The contents of the discharge atmosphere play a substantial role in the efficacy of NTP system; however, the information given across the publications diverges. Shi and Ileleji [62] have expressed that modified atmosphere gas containing 65% O_2_, 30% CO_2_, and 5% N_2_ (MA65) had a better degradation impact against aflatoxins in corn in comparison with air containing 78% N_2_ and 22% O_2_ under analogue conditions. The residual aflatoxins in corn decreased from 420 ± 21 ppb to 102 ± 17 and 161 ± 15 ppb after one min treatment in MA65 and air at 40% relative humidity, respectively. The authors have suggested the higher amount of reactive species as the main reason; the produced levels of NO_x_ and ozone were 10 and 3 times higher in MA65 than in the air after 5 min of NTP treatment. Additionally, the other results [63] showed that the increase in relative humidity from 5 to 40% leads to higher efficacy reaching 143 ± 24 and 102 ± 17 ppb residues, respectively, within 1 min of treatment. Probably, the increase in relative humidity acts in favor of generating higher concentration of OH radicals via reaction of ozone with H_2_O molecules [63]. Hydroxyl radical is a potent oxidizer with stronger activity than ozone [64] causing higher mycotoxin degradation than ozone.

However, the gas composition together with the discharge mode and character may play the crucial role. It is generally known that plasma-chemical reaction depends, among others, on the electron temperature. One example for all [69], the production of nitrogen oxides prevails over the initial ozone production for higher discharge intensities, which is known as a discharge poisoning effect. This may also be the explanation of seemingly contradictory results of Siciliano and Spadaro [58], where the addition of 21% of oxygen to the pure nitrogen completely dissolves the effect of NTP.

Hence, the balanced parameters are required for effective plasma treatment. e.g., in the work by Devi and Thirumdas [65], the increase in aflatoxin B1 reduction from 70 to 90% using higher discharge power (60 W instead of 40 W) was observed. So, the conditions differ from case to case and cannot be summarized in a single conclusion.

The last phenomenon is the increase in efficiency by extending the exposure time, as was reported in almost all studies. The interesting saturation effect was reported in several studies, where the reduction rates decreased at long exposure times, although the total mycotoxins degradation was not reached [66,67].

## 4. Electron Beam (EB) Radiation

The next technique is the EB involving the usage of accelerated electrons to treat a medium or object for a variety of goals [70,71]. This technology has wide application in many areas [72,73,74] as development of a new nano-material [75], novel metals [75,76,77,78], modification of carbon coatings [79], or decontamination in food industries [80]. Moreover, it has likewise found its path to agriculture and food systems, where the EB irradiation, as one of the ionizing radiations, can decay the structure of mycotoxins by its collision with the high energy electrons, or by the reaction with the secondary products of high-energy electrons generated mainly through the water ionization.

The usage of other sources of ionizing radiation for mycotoxin decontamination seemed to be a hopeful application in the past. In spite of many advantages such as, e.g., short processing time, in-line processing, more convenient irradiation, safer radiation, or lower costs than gamma irradiation [81,82,83,84,85], EB should be used carefully if applied on plants and food, since it may adversely affect their quality. It is due to the produced electrons, which can develop the oxidation processes or can change the organoleptic or technical properties of the foodstuff [86].

EB consists of a beam of highly energetic electrons, typically accelerated by a high electric field, and achieves the kinetic energy up to several MeV. It may be distinguished as a high- and low-energy beam for kinetic energy higher and lower than 300 keV, respectively. Depending on the energy, electrons penetrate from several micrometers up to several centimeters into the product [87]. As stated by Luo and Liu [6], the energy needed to breakdown the structure of mycotoxins is 5–10 MeV. A schematic of EB function for mycotoxin destruction is given in Figure 4.

### 4.1. Mechanism of Action of Electron Beam against Mycotoxins

The action mechanism of EB mycotoxins degradation can be provided in the direct and indirect way. In direct action, the electrons from EB collide with organic materials, leading to the destruction of their structure. In indirect action, reactive species generated in water participate in mycotoxin degradation. As reported by Hasanpour and Rahimi [88], the EB radiation of water leads to the radiolysis producing reactive species, especially H^•^ and OH^•^, along with secondary water electrons e-aq. Similar processes also occur in the moisture content of the food matrix, and generated radicals play the main role in the mycotoxins’ decay.

Peng and Ding [89] have proposed three pathways for decomposing of ochratoxin A in an aqueous solution by EB leading to six products (A–F), see Figure 5. The first way is the substitution of chlorine on the benzene ring by H^•^, generating the “F” product. The second way is the oxidation of ochratoxin by OH^•^ to produce D. The third way is the breakdown of the NH-CO bond by producing the “A” together with an intermediate structure. The “A” is hydrogenated by H^•^ forming the product “C”. The intermediate structure is oxidized by OH^•^ forming “E”. This “E” product is then dechlorinated and hydrogenated by H^•^ producing “B”. The decrease of 99.34% in ochratoxin A using 10 kGy dose of EB was also reported in this study.

Although the pathways and the generated products suggested by other researchers are different, probably due to different EB conditions and mycotoxins [90,91], they all agree on the role of H and OH radicals in degradation.

### 4.2. Effective Parameters of Electron Beam on Degradation of Mycotoxins

The most important parameters investigated in the recent studies are related mainly to the conditions of EB irradiation, including the effect of radiation dose, the initial concentration of mycotoxin, the moisture content, H_2_O_2_ amount, and the pH value. The influence of EB on different kinds of mycotoxins, matrices, storage, and processing are explored insufficiently; therefore, further studies in the future are required. Some of the mentioned parameters are discussed below.

#### 4.2.1. Irradiation Dose

Irradiation dose is a criterion of energy density deposited in mycotoxins. Nearly all the studies show a direct relation between the EB dose and mycotoxin degradation, though this relation is not necessarily linear. Luo and Qi [86] have indicated that the increase in the EB irradiation dose from 0 to 50 kGy enhanced the degradation rate of zearalenone and ochratoxin A in a corn grain from 0 to 71 and 68%, respectively. They likewise observed that this level of the increase was not the same for a corn grain. The higher degradation rate in a corn grain compared to a corn grain was attributed to the more distribution of mycotoxins on the surface, where it was accessible for the irradiation easily. A similar result was obtained by Assuncao and Reis [92] in the study of effects of the EB irradiation on aflatoxin on a Brazilian nut. They observed the decrease in aflatoxin amount from 4.75 to 2.21 and 1.63 µg/kg, when exposed to a 5 and 10 kGy dose, respectively.

The character of a matrix has a crucial role on the impact of EB dose. Some solvents have a radical scavenging effect, thus reducing the effect of EB radiation. For example, it was shown that the EB dose increase from 0 to 6 kGy gives different results in methanol and in acetonitrile solution both containing zearalenone and ochratoxin A [93]. The rate of mycotoxin degradation was significantly lower in the methanol solution. It has been suggested that methanol is a free radical scavenger [94]; therefore, radicals generated by EB in methanol were scavenged by methanol molecules. It was concluded that solutions with radical scavenging ability need a higher EB dose.

However, EB remains a promising technology to degrade mycotoxins, although it may also have adverse effects on the quality of edible materials, especially by increasing the irradiation dose. In this concern, a decrease in amylose content, essential and total amino acid contents, and starch crystallinity has been observed in treated corns by the EB dose increase from 10 to 30 kGy [93]. Electrons produced by EB may penetrate to deeper sections of matrix, which can result in changes in the color quality via the splitting of carotenoids and oxidation of lipids, the rancidity via producing free fatty acids, variation in the viscosity via degradation of starch, and some alteration in sensory properties of foods [86,91].

#### 4.2.2. Initial Concentration of Mycotoxin and Moisture

There is a conjecture about a direct relation between mycotoxin initial concentration and the primary degradation rate. It states that the increase in the initial concentrations leads to an exposition of a greater number of mycotoxins to the radiation, which results in a higher rate of degradation. Liu and Wang [91] showed that the aflatoxin B1 degradation rate at 2 kGy dose of EB radiation reduced with more intensity in aqueous mediums containing 5, 1, and 0.5 ppm of aflatoxin, respectively; however, the higher doses of EB are required to reduce the higher mycotoxin concentration to a specific level. There is no satisfying explanation of this situation in these studies. Maybe we can imply, in accordance with their studies, that the lower mycotoxin concentration has caused the higher surface of toxins to be exposed to EB radiation, hence improving the degradation yield. For example, Peng and Ding [89] found out that the irradiation doses to achieve 90% decay of ochratoxin should be nearly 2.19, 3.93, and 6.66 kGy at primary concentrations of 0.1, 0.2, and 20 mg/L, respectively.

The presence of water in the sample matrix leads to the occurrence of a higher number of radicals produced by the EB radiation, enhancing the degradation of mycotoxins. It has been shown in the work of Liu and Lu [95] that the degradation of aflatoxin B1 in peanut meals was significantly higher for the moisture content of 21%, than for 14 and 9%. Another important factor is the type of a solvent. In the study by Peng and Ding [89], it was shown that the degradation power of EB irradiation for ochratoxin dissolved in water is higher than in acetonitrile or methanol–water mixture. Authors suggest that the generated radicals may be scavenged in the organic solvents.

## 5. Pulsed Light

The last presented technique is the PL referring to the powerful short-time pulses of a broad-spectrum light [96]. This technology has a wide application in areas such as skin treatment [97], microbial inactivation on surfaces [98], and food sterilization [52,99].

Usually, the quantity of pulses of the light needed in a treating process is lower than 10, and every pulse can deliver an energy from 0.01 to 50 J/cm^−2^. Numerous pulses of light are delivered in a second, making this technique faster than the conventional methods for degradation, sterilization, or decontamination [53]. Additionally, PL treatment has been proven to be a suitable method, which does not change the food quality. Better maintenance of phenolic compounds and vitamin C and delayed browning and oxidation on fresh-cut apples treated by PL only support this claim. Similar results can be found in the study [54].

According to [55], the first works on disinfection with flash lamps date back to the late 1970s in Japan. Bank, John, Schmehl, and Dracht seemed to be the first researchers who have published scientific articles about the application of PL to inactivate microorganisms. By using a UV-C light source of 40 W at the maximum peak power, a 6–7 log decrease in viable cell numbers was achieved [100].

Moreover, PL is also a novel, FDA-approved, non-thermal technology with the potential to degrade mycotoxins in solutions, food, and their by-products [53,101,102]. For instance, it achieved a 84% degradation of zearalenone was reported by Morea, using only 8 flashes of PL in wavelengths from ultra-wave to near-infrared, from 180 to 1100 nm.

PL is generated by means of engineering technologies that amplify the power several times to transform high-speed electronic pulses to short-duration and high energy light pulses. The system consists of three main components: the power supply, the pulse configuration instrument, and the lamp (Figure 6). First, the energy is stockpiled in a high-power capacitor for a moderately long period of time (a fraction of a second); then, the stored light is delivered to a particularly proposed xenon lamp unit in a much shorter time (nanoseconds to milliseconds), producing an intense and a few hundred microseconds short pulse of light focused on the treatment area [99]. The light produced by the lamp comprises a broad spectrum of wavelengths from UV to near-infrared (180–1100 nm) [103].

### 5.1. Inactivation Mechanism of Pulsed Light against Mycotoxins

The mycotoxin degradation effects of PL can be attributed to its UV content and a high peak power. However, there is a limitation of PL given by the transparency of the sample. For a transparent matrix, the light penetrates deeper and allows for complete decontamination. Unlike for an opaque matrix, the effect of PL is limited only to the first 2 μm of the surface [104].

Published articles have shown that two mechanisms, the photochemical and the photothermal ones, are involved in general [105,106] (Wang et al., 2016, Mandal et al., 2020). Both mechanisms coexist, but the relative importance of each one probably depends on the targeted mycotoxin. Photochemical mechanism of PL is attributed to its UV-C spectrum, where the intense short duration flashes with high peak power lead to the photochemical breakdown of mycotoxins. Although PL is considered as a non-thermal method, the photochemical mechanism is attributed to local short time heating of mycotoxins leading to their photothermal destruction.

The work of Wang et al. [105] reported that PL intensity exceeding 0.5 J/cm^−2^ leads to the degradation through mycotoxins disruption during their temporary overheating from the absorption of all UV light. This overheating can be attributed to a difference in UV light absorption by mycotoxins and that of surrounding media.

### 5.2. Parameters Affecting the Pulsed Light Efficiency

Several studies have shown that the efficiency depends on various factors such as the power intensity, the exposure time, mycotoxin concentrations, and a sample character. A summary of these parameters is discussed below. No extensive research has been conducted on this issue, and the reported results show high variability.

#### 5.2.1. Mycotoxin Concentration

In the study conducted by Funes and Gómez [107] the effect of PL dose was investigated on patulin degradation of McIlvaine buffer, apple juice, and apple puree. The exposure of all samples to PL doses between 2.4 and 35.8 J/cm^2^ resulted in a significant decrease in patulin levels. However, patulin reduction in McIlvaine buffer did not depend markedly on the initial concentration of the mycotoxin. As an opposite result, let us mention a work of Wang and Mahoney [108] who reported that the degradation rate of AFB1 (230, 31, and 18 μg kg^−1^) and AFB2 (248, 32, and 20 μg kg^−1^) in a solid medium was proportional to the initial concentrations of the aflatoxins.

Different results in the effect of initial toxin concentration on the efficacy of PL treatment may be associated with the difference in the type of medium. It has been suggested that in a non-solid media, there is a reverse or independent relation between a toxin concentration and a PL treatment [108]. Commonly, the photoreaction rate under a constant temperature mostly depends on the number of reactants, so that it increases with the increasing number of reactants. Based on the collision theory, the photoreaction rate directly relates to the number of successful collisions in each second between reactants. Nevertheless, the reaction medium affects the rate of reaction. Jablo stated that in a liquid environment, UV can generate active elements such as hydrogen peroxide, single state oxygen, hydrated electrons, hydroxyl radicals, and peroxy radicals that elevate the degradation rate. Jing et al. [109] clarified that the quantity of reactive elements produced from UV irradiation was sufficient to activate the photoreactions when the primary concentration of naphthalene in water was low, while utilizing the same number of active elements may not be enough to achieve the reaction in a favorite rate if the primary amount of toxin is much higher. Therefore, in a non-solid medium, the photodegradation rate may display a reverse or even an independent relation to the primary concentration of toxin. On the other hand, in case of solid media, not sufficient amounts of active particles are generated to affect the degradation rate.

#### 5.2.2. Number and Exposure Time of Pulses

The number and exposure time of pulses are critical parameters for process optimization, in order to maximize the efficiency against mycotoxins and to minimize the product damage. Changes in food traits can be attributed to thermal damage. Abuagela and Iqdiam [110] investigated the effect of PL treatment in a radiant exposure of 4, 8, 12, and 16 J.cm^−2^ with the frequency of three pulses per second in conjugation with a pulse length of 360 μs on the reduction in aflatoxins (B1 and B2) on contaminated peanuts (with and without skin). The best reduction in AFT was 82 and 91% for with-skin and without-skin peanuts was observed after 5 min. No significant differences (*p* > 0.05) were observed for all peanut oil quality parameters, which indicates that PL can be applied as a decontamination system with minimal destructive effect on a food quality.

One of the most substantial parameters affecting the time of PL treatment is the type and texture of samples. Wang and Mahoney [105] examined the efficiency of pulsed light treatment for degradation of aflatoxin B1 and B2 in rough rice and rice bran. In this examination, rough rice was first inoculated with Aspergillus flavus to produce AFB1 and AFB2, followed by PL treatments of 0.52 J/cm^2^/pulse for various duration times. The PL treatment time of 80 s reduced AFB1 and AFB2 in rough rice by 75 and 40%, respectively; while a shorted treatment time of 15 s reduced AFB1 and AFB2 in rice bran by 90 and 87%, respectively.

In another study [54], a pilot-scale PL applicator was used to treat 5 mL samples of a peanut oil of 10 mm thick. Results showed 48, 56, and 78% aflatoxin reduction for 400, 600, and 800 s exposure times, respectively.

## 6. Comparison of Presented Methods

The summary of the main properties of presented non-thermal methods are presented in Table 2. All methods are currently available in laboratory scale only; however, from the principle of action, the estimated cost of final apparatuses is not high. The efficacy of mycotoxins removal in in vitro experiments is reaching 100% and 70–90% in real food samples depending on specific mycotoxin structure, food matrix, or presence of possible antioxidants. While NTP acts mainly on the surface of the sample, EB can penetrate the whole one and PL depends on its transparency. Possible known undesirable effects, which have to be considered, are the oxidation of samples by NTP, and the local overheating by LP; for EB, there are currently no reported effects; however, it is due to the lack of available studies. However, from the industrial point of view, EB and especially NTP have developed in the food industrial scales in recent years [36,80], most use of them in the food industry are based on ameliorating the properties of polymers. 

## 7. Conclusions

NTP, EB irradiation, and PL are modern technologies capable of mycotoxin degradation with high efficiency and potential practical applications.

For NTP, the total degradation of pure mycotoxins and up to 70% degradation on real food samples in several minutes may be achieved under sufficient conditions. The presence of oxygen and humidity in the atmosphere was considered as one of the main parameters affecting the efficiency.

For EB, the results and mechanisms are similar to NTP. In this case, more than 99% efficiency was achieved under sufficient conditions. In general, the efficiency increases in the presence of water in irradiated samples and decreases by the presence of organic solvents, which is probably caused by the scavenging effect.

For the PL, the degradation of mycotoxins is more than 90%. The degradation is mainly induced by the UV light; however, it was also observed by a visible and a near infrared light. The photochemical and photothermal mechanisms are suggested, and the efficiency strictly depends on the sample transparency.

A direct comparison of efficacy of NTP, EB, and PL on the mycotoxin degradation is a complicated issue, since there are many factors that should be addressed before, including the food matrix and the type of mycotoxin and, most importantly, the intensity of process used in each technique. Therefore, there is a need for studies, comparing developing methods against mycotoxins in the same conditions. Nevertheless, it seems that the application of EB and NTP against food mycotoxins in the industry is in the developing stages. PL technology due to its restriction in capacity of treatment has not been employed in large-scale application yet. Hence, it appears that EB and NTP are preferred compared to PL to be used in food industries.

## Figures and Tables

**Figure 1 jof-07-00395-f001:**
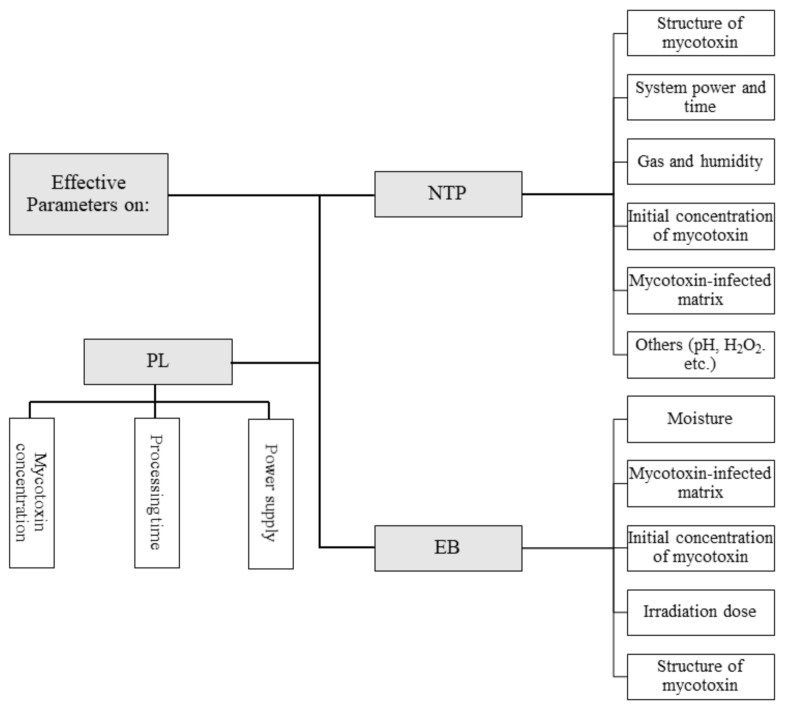
The outline of most important parameters of presented non-thermal methods (EB—electron beam, NTP—non-thermal plasma, PL—pulsed light) with the respect to the efficiency against mycotoxin removal investigated among studies. The mycotoxin concentration, processing time, or dose are the common parameters for all methods. Moreover, each method has its own specific parameters.

**Figure 2 jof-07-00395-f002:**
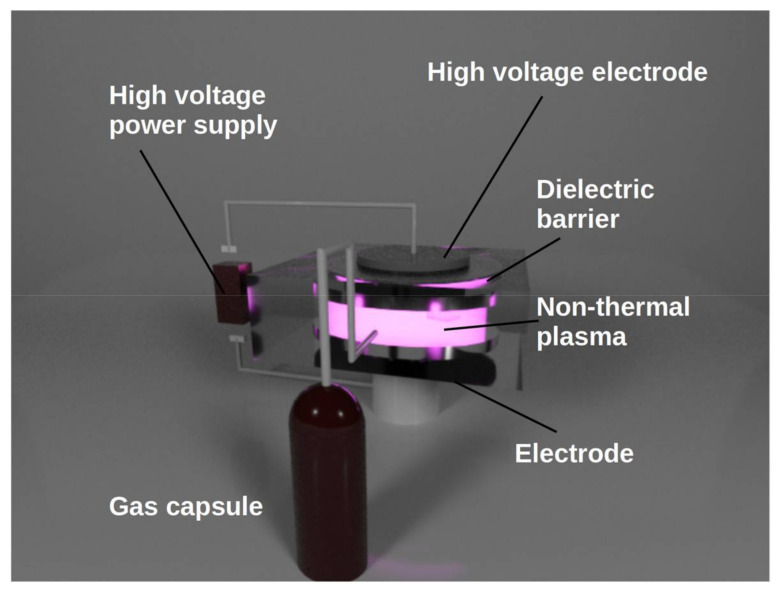
A schematic of dielectric barrier discharge.

**Figure 3 jof-07-00395-f003:**
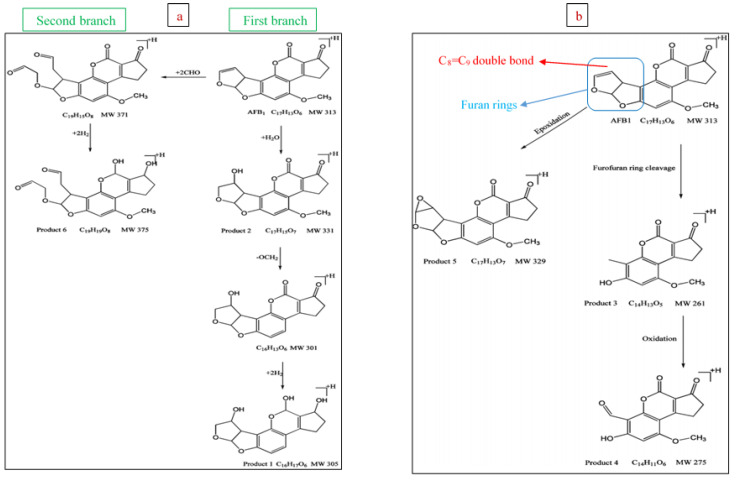
Degradation pathways (**a**,**b**) of aflatoxin B1 using non-thermal plasma.

**Figure 4 jof-07-00395-f004:**
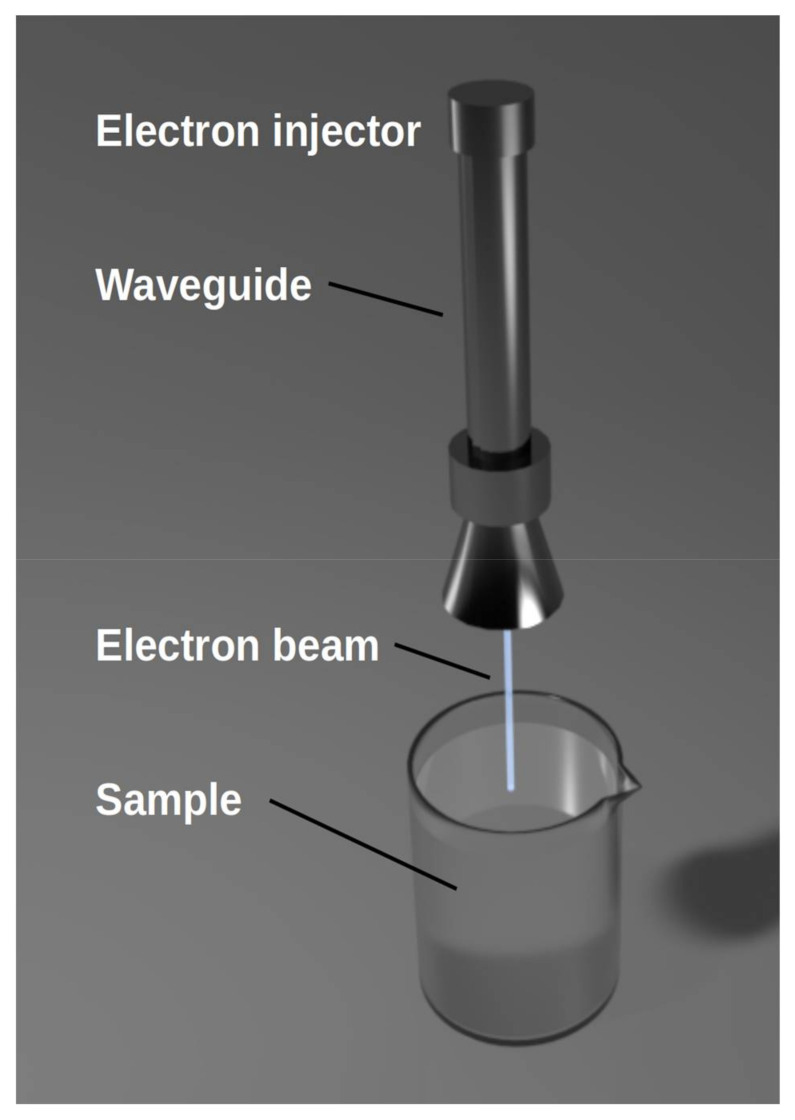
Electron beam function against mycotoxins.

**Figure 5 jof-07-00395-f005:**
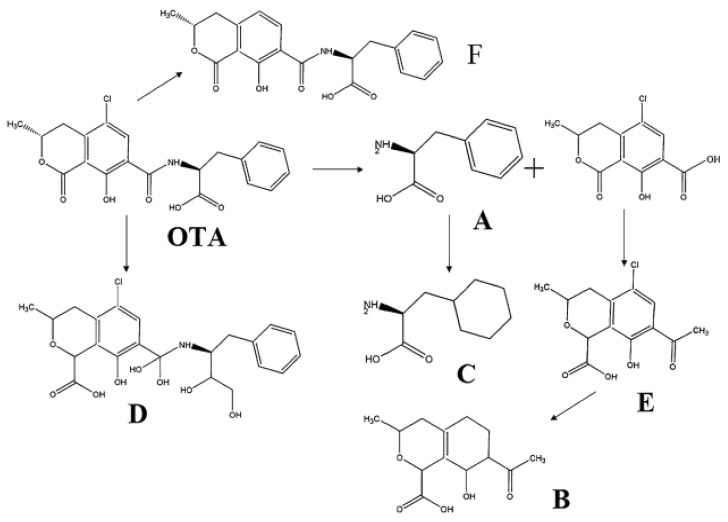
Degradation pathways of ochratoxin A (OTA) into six fragments using electron beam irradiation.

**Figure 6 jof-07-00395-f006:**
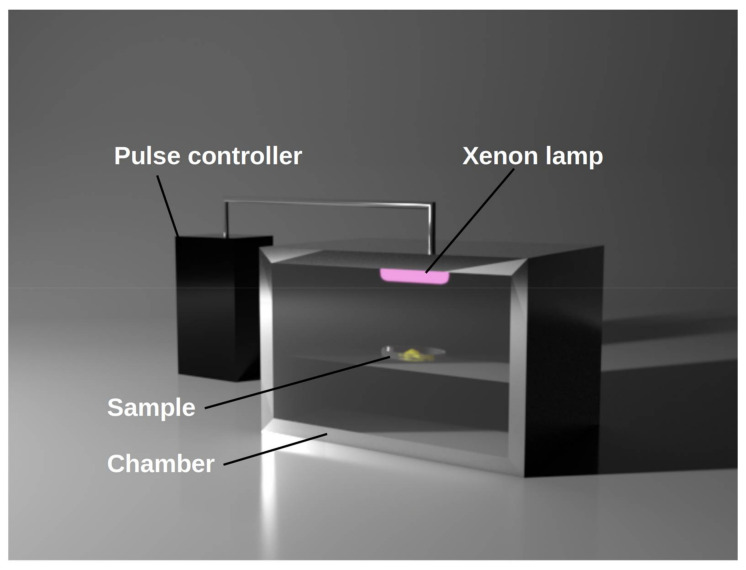
A schematic of pulsed light system.

**Table 1 jof-07-00395-t001:** Conditions applied for mycotoxin decontamination from some food samples using non-thermal plasma.

Generating System	Gas	Voltage (kV)	Frequency (kHz)	Power (W)	Time (min)	Contaminated Sample	Mycotoxin	Mycotoxin Reduction (%)	Reference
Dielectric barrier discharge	Helium	0.85	-	30	30	Roasted coffee	Ochratoxin	50	[52]
Corona discharge	Air	20	58	-	30	Rice and wheat	Aflatoxin B_1_	45–56	[53]
Atmospheric pressure plasma jet	Air	-	25	655	1.7	Hazelnut	Aflatoxin B_1_ and B_2_	70–71%	[54]
Dielectric barrier discharge	Helium and oxygen	6	20	-	10	Maize	Aflatoxin B_1_ and fumonisin B_1_	66	[55]

**Table 2 jof-07-00395-t002:** Comparison of specific methods with the respect to efficacy, limitations, and undesirable effects.

	NTP	EB	PL
**Generated by**	electrical discharges	high electric field, kinetic energy of electrons up to several MeV	flash lamps, from IR to UV
**Mechanisms**	generated reactive species	electron collisions the mycotoxin structure reactive species generated in water	powerful short-time pulses of a broad-spectrum light UV content local overheating by high peak power
**Efficacy**	up to 100% in vitro up to 70% in food long aliphatic chains have less resistance as aromatic rings	up to 99% in water solution up to 70% in food lower mycotoxin concentration caused the higher exposed surface of toxins presence of water leads to higher number of radicals produced generated free radicals are scavenged by some solvents, e.g., methanol	up to 90%
**Limitations**	laboratory scales only surface action only antioxidant substances decreases the effect	laboratory scales only penetration from several micrometers up to several centimeters depending on the energy	laboratory scales only deeper penetration for a transparent matrix allowing complete decontamination for opaque matrix, the effect is limited to the surface
**Undesirable effects**	oxidation of matrix	insufficiently explored	overheating of matrix

## Data Availability

Not applicable.

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
