# Peer review of "Application of Novel Non-Thermal Physical Technologies to Degrade Mycotoxins"

_jof, 2021, doi:10.3390/jof7050395_

Round 1

Reviewer 1 Report

The harmful effect of mycotoxins on health is unquestionable and therefore research in new methods of degradation of mycotoxin contamination from agriculture products and food is very important. This paper brings an overview of previous knowledge about possibility of physical, nonthermal technologies applicatoin, which can replace chemical, biological or thermal treatment, often accompanied by drawbacks.

 Based on the literature review, the relationships between the chemical structure and the degradation mechanisms of selected types of mycotoxins depending on the kind and processing conditions are formulated.

Although these are three different types of technology, (non-thermal plasma, electron beam, and pulsed-light treatment), and the individual studies work with different variables, the question is whether the findings on the effectiveness of mycotoxin degradation in food could not be summarized and generalized together.

I suggest this paper for publication after folloving  little revisions (comments  referring to line numbers):

42 ... on the quality of foods and plants is, in general, only negligible.

It would be better to write a less unambiguous statement. Impact of these technologies on food quality may not always be negligible, even when compared to other means of disposal of mycotoxins is probably minimal.

169 -171 ...., but this time for inoculated extract of rice with the same amount (100 μg mL-1) of fumonisin B1 was measured 60 seconds.

What is the depth of penetration?

187 – 188 The heat (<60 °C) and UV intensity  (50 μW/cm) produced during the generation of non-thermal plasma is far less than needed for mycotoxins degradation.

What UV intensity is needed for mycotoxins degradation?

337 - The influence of NTP on different kinds of mycotoxins, matrices, storage and processing are explored insufficiently, therefore further studies in the future are required.

Here should be noted influence of EB instead of NTP.

It would be good to unify the terminology related to the experimental material. There are all possible terms for grain, flour, etc. used in the text.

Author Response

Authors would like to thank both the reviewers for pertinent remarks helping to improve our manuscript. Our responses to the comments are:

  • 42 ... on the quality of foods and plants is, in general, only negligible.
    It would be better to write a less unambiguous statement. Impact of these technologies on food quality may not always be negligible, even when compared to other means of disposal of mycotoxins is probably minimal.
    We used the term “may be” despite of “is”
  • 169 -171 ...., but this time for inoculated extract of rice with the same amount (100 μg mL-1) of fumonisin B1 was measured 60 seconds.
    What is the depth of penetration?
    This is not specified in the paper. We can calculate, the thickness of inoculated extract was cca 0.1 mm, however, we are unable to find the thickness of extract after evaporation. We can estimate 10 μm. We have add this data to the manuscript.
  • 187 – 188 The heat (<60 °C) and UV intensity  (50 μW/cm) produced during the generation of non-thermal plasma is far less than needed for mycotoxins degradation.
    What UV intensity is needed for mycotoxins degradation?
    There is no strict limit, but for example the intensity of 800 μW cm‐2 is needed to successfully degrade the AFB1 from peanut oil. We have add this explanation with reference to the text.
  • 337 - The influence of NTP on different kinds of mycotoxins, matrices, storage and processing are explored insufficiently, therefore further studies in the future are required.
    Here should be noted influence of EB instead of NTP.
    Yes, thank you, corrected.
  • It would be good to unify the terminology related to the experimental material. There are all possible terms for grain, flour, etc. used in the text.
    We have tried to unify all possible terminology used in the manuscript.

Reviewer 2 Report

Dear authors,
Your article discussed a very relevant issue. In my expert opinion, your manuscript is already excellent. However, please consider the observations below before proceeding with the publication process:

ABSTRACT
Your abstract is insufficient. The abstract should be a miniature of your manuscript, and it should be able to provide a complete overview of your work for people who do not have time to read it. Your key findings and conclusion should be there. I recommend you look at other abstracts and follow the same logic.

1. INTRODUCTION
Line 27: remove "which are mainly".
Figure 1: the legend should be more explanatory about the relationship between the methods and the parameters mentioned.

2. MYCOTOXINS
Line 52: "Mycotoxins are secondary metabolites produced by some filamentous fungi" - redundant. You wrote this in the introduction.

Lines 64-65: "many of them are classified as mutagenic, carcinogenic, or genotoxic" - redundant.

Lines 65-66: "more than 400 compounds identified as mycotoxins" - repetition. Avoid them by being brief and direct.

Line 73-86: could be converted into a table.

Be consistent in your flow of ideas. For instance, you can write a definition and description of aflatoxins, what they cause, where to find and how to prevent them. Avoid going back and forth.

Line 107: review the parentheses and punctuation

3. NON-THERMAL PLASMA
Line 115: you already explained in the introduction that NTP stands for non-thermal plasma. From that point on, use the abbreviation.

Lines 235 and 253: is it heading 1, 2 or 3?

CONCLUSION
Elaborate more and explain which do you consider better after a thorough evaluation of cost-benefit.
Leave your opinion and recommendations.

MORE OBSERVATIONS
I will not nitpick so much. I would like to see:
1. A section before the conclusion where you compare the methods considering the efficacy, potential undesirable effects on food, cost, etc. You can use a table for this purpose.
2. For each method, discuss more their affordability and who will potentially use them.

3. Are you the author of the pictures. If yes, you do not need to change anything. If not, please mention the sources and their licenses.

Yours sincerely

Author Response

Authors would like to thank both the reviewers for pertinent remarks helping to improve our manuscript. Our responses to the comments are:

  • ABSTRACT Your abstract is insufficient. The abstract should be a miniature of your manuscript, and it should be able to provide a complete overview of your work for people who do not have time to read it. Your key findings and conclusion should be there. I recommend you look at other abstracts and follow the same logic.
    We agree, we have add several sentences in the Abstract.
  • 1. INTRODUCTION
    • Line 27: remove "which are mainly".
      We agree, removed.
  • Figure 1: the legend should be more explanatory about the relationship between the methods and the parameters mentioned.
    The legend was enhanced.
  • 2. MYCOTOXINS
    • Line 52: "Mycotoxins are secondary metabolites produced by some filamentous fungi" - redundant. You wrote this in the introduction.
      The redundancy was removed
    • Lines 64-65: "many of them are classified as mutagenic, carcinogenic, or genotoxic" - redundant.
      The redundancy was closed to brackets.
    • Lines 65-66: "more than 400 compounds identified as mycotoxins" - repetition. Avoid them by being brief and direct.
      The redundancy was removed
    • Line 73-86: could be converted into a table.
      We agree the list is unnoticed, however we do not consider to insert table in the Introduction as suitable, we have formatted the text for better readability.
    • Be consistent in your flow of ideas. For instance, you can write a definition and description of aflatoxins, what they cause, where to find and how to prevent them. Avoid going back and forth.
      WeI agree, we have rewrite several sentences
    • Line 107: review the parentheses and punctuation
      Corrected.
  • 3. NON-THERMAL PLASMA
    • Line 115: you already explained in the introduction that NTP stands for non-thermal plasma. From that point on, use the abbreviation.
      Corrected.
    • Lines 235 and 253: is it heading 1, 2 or 3?
      It is heading 3, in the manuscript template it was without the numbering. The numbering was added (as 3.3.1 and 3.3.2). Similar changes was done in 4.2.1, 4.2.2, 5.2.1 and 5.2.2.
  • CONCLUSION Elaborate more and explain which do you consider better after a thorough evaluation of cost-benefit. Leave your opinion and recommendations.
    We agree, we have add several sentences into the Conclusion.
  • MORE OBSERVATIONS I will not nitpick so much. I would like to see:
    • 1. A section before the conclusion where you compare the methods considering the efficacy, potential undesirable effects on food, cost, etc. You can use a table for this purpose.
      The section and table were added.
    • 2. For each method, discuss more their affordability and who will potentially use them.
      We agree, we have add several sentences into the text.
    • 3. Are you the author of the pictures. If yes, you do not need to change anything. If not, please mention the sources and their licenses.
      Yes, we are the authors.